# TDCS over PPC or DLPFC does not improve visual working memory capacity
**Shuangke Jiang** [ID] ✉, **Myles Jones** [ID] & **Claudia C. von Bastian** [ID] ✉

Non-invasive brain stimulation has been highlighted as a possible intervention to induce cognitive benefits, including on visual working memory (VWM). However, findings are inconsistent, possibly due to methodological issues. A recent high-profile study by Wang et al.[1] reported that anodal transcranial direct current stimulation (tDCS) over posterior parietal cortex (PPC), but not over dorsolateral prefrontal cortex (DLPFC), selectively improved VWM capacity but not precision, especially at a high VWM load. Thus, in the current pre-registered conceptual replication study, we accounted for the key potential methodological issues in the original study and tested an adequate number of participants required to demonstrate the previously reported effects (n = 48 compared to n = 20). Participants underwent counterbalanced PPC, DLPFC and sham stimulation before completing 360 trials of a continuous orientation-reproduction task with a slight variation of task stimuli and setup. We found no evidence for the selective effect of PPC stimulation. Instead, our results showed that tDCS effects were absent regardless of stimulation region and VWM load, which was largely supported by substantial to strong Bayesian evidence. Therefore, our results challenge previously reported benefits of single-session anodal PPC-tDCS on VWM.

Visual working memory (VWM) refers to the active maintenance of visual information needed for higher cognitive processing in the present moment[2]. Typically, working memory (WM) is limited to maintaining three to four chunks of information[3]. Like other fluid cognitive abilities, WM declines with age[4,5]. Furthermore, deficits in WM often occur with neurological diseases and psychological disorders. The limited capacity of WM, and its critical involvement in many disorders, has stimulated intensive research efforts into the effectiveness of WM enhancement interventions.

In particular, there is growing interest in affordable and non-invasive brain stimulation techniques, such as transcranial direct current stimulation (tDCS). The benefits of tDCS have been demonstrated for healthy young adults[6–13], healthy older adults[14] as well as clinical cohorts with mild cognitive impairment and early Alzheimer's disease[15,16], attention-deficit hyperactivity disorder[17,18] and major depressive disorder[19,20].

Typically, tDCS delivers weak currents from anode to cathode through the skull, generating electric fields to modulate cortical activities and facilitate neuroplasticity[21,22]. Anodal stimulation is assumed to increase cortical excitability to enhance cognitive functions, whereas cathodal stimulation decreases excitability and, thus, inhibits brain activities[23,24]. However, such polarity-specific effects of tDCS are likely an oversimplification when considering complex cognitive functions like VWM. For example, whereas excitatory effects of anodal stimulation are largely robust, inhibitory effects of cathodal stimulations are less consistent when it comes to studies investigating complex cognition rather than motor skills[25]. Taken together, regardless of inconsistent cathodal effects, anodal stimulation has been shown to consistently modulate the neural activities in the target brain regions and, thus, is a promising avenue to enhance the corresponding cognitive functions.

Given that neural activation of frontal-parietal brain regions is known to be involved in the maintenance of VWM representations[26,27], a growing body of research has investigated the possible VWM benefits of anodal stimulation of the dorsolateral prefrontal cortex (DLPFC) and posterior parietal cortex (PPC)[1,28–30]. However, some previous studies have showed null effects of tDCS for both DLPFC and PPC stimulation[31–33]. Some meta-analyses quantifying the effectiveness of tDCS across multiple studies report medium effects of single-session tDCS on VWM[34–36], while others report only negligible effects of single-session tDCS[37–40]. These inconsistencies on the meta-analytic level point to several critical caveats of meta-analyses. Specifically, any conclusions drawn from meta-analysis depend on the included primary studies. First, if the included primary studies largely reported only positive effects, together with overestimated study-level effect sizes[41], it can lead to high false-positive rates of meta-analyses[42,43]. Second,

Department of Psychology and Neuroscience Institute, University of Sheffield, Sheffield, UK. ✉e-mail: jiangshuangke@gmail.com; c.c.vonbastian@sheffield.ac.uk

tDCS studies vary widely in their design, such as administering online or offline protocols[44], stimulating different regions[34], or using different VWM paradigms which may require different cognitive processes to one another[45]. These methodological variations could have contributed to the inconsistencies observed across both single studies and meta-analyses. Therefore, replications of those studies that reported positive results, using the same parameters and cognitive paradigms, may yield more conclusive evidence as to whether tDCS is effective or not.

The present, pre-registered replication study, therefore, focuses on a particularly high-profile study by Wang et al.[1] who recently reported selective benefits of anodal tDCS over the PPC, but not DLPFC, on VWM. Wang et al.[1] used a continuous-reproduction VWM paradigm and fitted the mathematical standard mixture-model[46] to estimate VWM capacity (quantity of representations maintained in VWM) and precision (quality of those representations). In this task, participants memorized the orientations of 2, 4, or 6 bars on a screen. After either a short (100 ms) or long (1000 ms) interval, participants were asked to reproduce the orientation of one of the bars by mouse-click. The deviation of the reproduced orientation from the original orientation was then used to estimate VWM capacity and precision for each participant, interval duration, set size, and stimulation condition. Wang et al. tested the effects of 15-min, 2 mA anodal tDCS over the left DLPFC and the right PPC relative to a sham condition with a within-subjects design in 20 participants. After excluding two participants due to their poor performance at set size 6, Wang et al.[1] observed a selective increase in VWM capacity for the long retention interval at this set size after PPC stimulation relative to sham, but not after DLPFC stimulation, at any other set size, short retention interval, or on VWM precision.

Wang et al.[1] interpreted these findings as "causal evidence" (p. 535) of the role of the PPC for VWM functioning. They further argued that "tDCS could be used as a promising noninvasive method to enhance [VWM]"[1] (p. 535). Indeed, Wang et al.'s[1] findings have several important theoretical and practical implications to the fields of VWM and tDCS. First, the findings from this study falsified the role of anodal stimulation at 2 mA on the DLPFC in improving VWM, thereby contradicting previous studies in which a weaker current (1 mA) and different WM paradigms (e.g., digits forward-span and 1 to 3-back) were administered[6,7,13,47]. Indeed, recent studies reported an overall absence of anodal DLPFC-tDCS effects on enhancing WM performance regardless of current intensity[33,48]. Furthermore, one consistent finding regarding differences in task type across several studies is that individuals benefit from tDCS when WM tasks are more demanding[10,48,49]. Second, by showing that tDCS selectively increases the capacity, but not the precision, of representations held in VWM, Wang et al.'s[1] findings strongly favor theories conceptualizing the capacity limit of VWM as discrete memory slots[46] over those assuming a flexible, continuous resource[50,51]. Third, the promising benefits of PPC-tDCS suggests that VWM capacity can be expanded with a non-invasive, cost-effective method, with strong practical implications for clinical tDCS applications. Importantly, by employing a sham-control and through the null effects of DLPFC stimulation, Wang et al.[1] excluded the possibility that these changes were driven by placebo effects or global excitability with tDCS[52]. Furthermore, Wang et al.'s[1] additional control for sensory memory also ruled out the possibility that these changes were due to mere sensory processes or attentional regulation.

Given these far-reaching implications, it is imperative to ensure that Wang et al.'s[1] findings are robust and replicable. Replication studies can verify the reliability of the originally reported effects[53], and test the generalizability across conditions that inevitably differ from the original study[54]. This is particularly critical in the present replication study because, despite its important findings and implications, several aspects of Wang et al.'s[1] study are potentially problematic and warrant further investigation. First, Wang et al.[1] retained only a small sample of 18 participants for analysis. The small sample size translates into low statistical power even for moderate effect sizes, and low statistical power can lead to false-positive findings[55]. The reported effect size is very large ($d = 1.028$), but this may reflect an overestimation due to low statistical power[41]. Second, Wang et al.[1] administered only 60 trials per design cell. Such relatively small numbers of trials increase

bias and noise variance and thus reduce the precision of the parameter estimation[56,57]. Third, the counterbalancing of conditions was likely incomplete in Wang et al.[1] and, thus, their design did not adequately control for possible carryover (e.g., practice) effects across sessions. Specifically, the study entailed three sessions (DLPFC, PPC, and sham stimulation), resulting in at least six possible sequences requiring counterbalancing. However, with 20 participants completing the experiment (and 18 included in the analysis), it is impossible to assign an equal number of participants to all sequences. Consequently, it cannot be excluded that carryover effects contributed to the previously reported effects. Finally, Wang et al.[1] used rotated bars as stimuli. The unique angles of their stimuli effectively ranged only from "10° to 170°" (p. 529), leaving room for developing task-specific strategies. For example, participants may have realized that simply memorizing the location of either end of the bar (90° or 270°) will result in the correct response (90°), thereby making the VWM task considerably easier than when presenting stimuli that use the full space of 360 possible responses.

To address these potential issues of their study, in this pre-registered experiment, we aimed to replicate Wang et al.'s[1] study, using a bigger sample size, larger number of trials, complete counterbalancing, and stimuli that use the full space of possible angles (0° to 359°). Our pre-registered hypotheses (https://osf.io/n9fkp) based on Wang et al.'s[1] findings were as follows:

Hypothesis 1: PPC stimulation will increase VWM capacity more than DLPFC stimulation. This effect is particularly pronounced at a high difficulty level of the task (i.e., set size 6).

Hypothesis 2: Neither PPC nor DLPFC stimulation will improve VWM precision.

## Methods

This experiment and our hypotheses were pre-registered on the Open Science Framework (https://osf.io/n9fkp) on March 12, 2020. A pilot study served to test the feasibility of the study, the safety of current tDCS setup and the feasible workflow of the analysis. These pilot data were not included in the analyses of the present study. The study was approved by the University of Sheffield Research Ethics Committee.

### Participants

A total of 48 healthy young adults were recruited (31 female participants, 17 male participants; all right-handed; $M$ and SD of age 22.65 ± 4.34 years, range 18–33 years). All participants were retained for analysis. We chose this sample size for two reasons. First, although Wang et al.[1] reported a large effect size of Cohen's $d = 1.028$, yielding a (post-hoc) power of $1 - \beta = 0.98$ for their included sample of 18 participants for analysis, simulations have shown that effect sizes are often overestimated for such small samples[41]. Therefore, we ran an *a priori* power analysis based on a more conservative medium effect size of Cohen's $d = 0.50$, a power of $1 - \beta = 0.90$ and an α-level of 0.05, resulting in a minimum sample size of 44 (G*power 3.1[58]). Second, fully counterbalancing the stimulation conditions (i.e., DLPFC, PPC and PPC/DLPFC sham) across three sessions results in 12 possible combinations; therefore, we recruited 48 healthy participants, which is a multiple of 12 and more than twice larger than the sample size in the original study.

The inclusion criteria were similar to those in Wang et al.[1]: all participants had normal or corrected-to-normal vision, no metallic implant, and no history of any neurological or psychiatric illness. In addition, in the present study, only participants who were proficient in English and educated to A-level or higher were included. Furthermore, we excluded participants who self-reported that they underwent neurostimulation within the past week, were on medication with known cognitive side-effects, in particular on memory and attention, or were currently using recreational drugs (e.g., cannabis, cocaine, or methamphetamines), or were pregnant. Participants were recruited through university volunteer systems, social media (e.g., Facebook), display of flyers, and word-of-mouth. Participants were compensated with £15 or £5 and course credits.

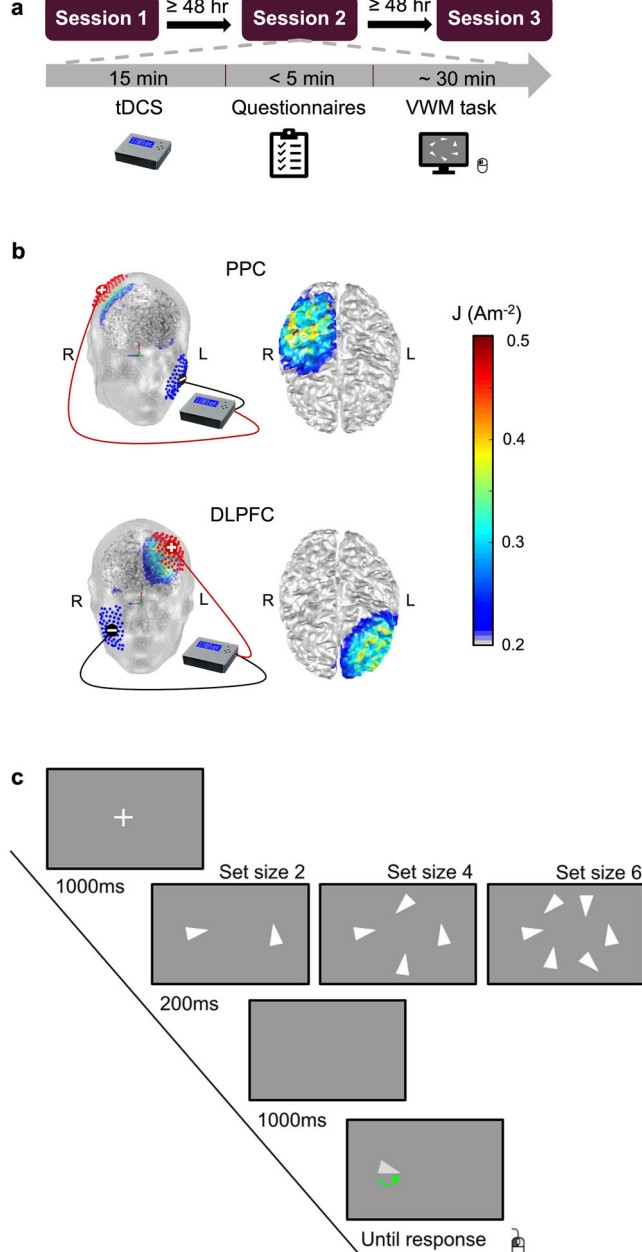

**Fig. 1 | Study Overview. a** Study procedure. **b** TDCS montages on head models (left) and current density distributions from the superior view (right). Red electrodes with a cross: anode; black electrodes with a line: cathode. **c** Continuous orientation-reproduction task. PPC right posterior parietal cortex, DLPFC left dorsolateral prefrontal cortex, L left, R right.

## Procedure

Figure 1 provides an overview of the study procedure, tDCS setup, and VWM task used. During each experimental session, participants first received either active PPC stimulation, active DLPFC stimulation or sham. Following short questionnaires, post-tDCS VWM performance was measured using a continuous orientation-reproduction task. During each trial of the VWM task, participants memorized the orientations of 2, 4, or 6 triangles on a screen. In this replication, we included only the relevant maintenance condition with a long (1000 ms) retention interval, after which participants were asked to reproduce the orientation of one of the triangles. The deviation of the reproduced orientation from the original orientation was then used to estimate VWM capacity and precision for each participant, set size, and stimulation condition.

This lab-based study used a within-subjects, randomized and single-blinded design (Fig. 1a). All participants came to the lab for three sessions. Each session lasted about 1 h, with an intersession-interval of at least 48 h to allow for any possible after-effects of tDCS to return to baseline ('wash out'). Upon arrival at their first session, participants gave their written informed consent for their participation and completed a self-report questionnaire on demographic information (age, sex, main language, handedness, and education level). At each session, participants first received the tDCS. Next, they completed short post-stimulation ratings (see Supplementary Notes 1) on their current pain, attention, and fatigue levels, followed by a tDCS adverse-effects questionnaire (see Supplementary Notes 2 and Supplementary Table 1). Next, participants completed a computerized VWM task. In addition, to measure expectation effects, at the end of the third and final session, participants were asked to guess whether they had received active or sham stimulation at each session (see Supplementary Notes 1). Note that these data are not reported in this article, but results from analyses of these data can be found in the Supplementary Notes 1 and Supplementary Notes 2. Overall, the current tDCS setup did not largely lead to any severe adverse effect, which indicates the safety of the montage and paradigm. The pattern of results confirms that sham stimulation provided a good level of condition blindness, thereby preventing placebo effects.

### TDCS setup

A battery-driven TCT Research tDCS 1ch device was used to deliver direct current via two saline-solution-soaked sponge electrodes (electrodes size: $5 \times 7$ cm$^2$; https://trans-cranial.com). Figure 1b illustrates the current density model for the two active stimulation conditions simulated by MATLAB-based COMETS toolbox[59]. Identical to the unilateral stimulation in the original study, the anodal electrode was placed at the target stimulation brain regions, that is, either the left DLPFC or the right PPC, while the cathodal electrode was placed on the contralateral cheek. In each session, participants received one of the three types of stimulation (active DLPFC, active PPC, or sham) for 15 min. The order of the three stimulation conditions was counterbalanced across participants.

For half of the participants, sham stimulation was on the left DLPFC, and for the other half sham stimulation was on the right PPC. In the active stimulations, the tDCS current linearly reached 2 mA within the first 30 s (20 s in the original study) and then remained stable until the last 2 s (20 s in the original study) when the current gradually decreased until tDCS was turned off. The sham stimulation followed the same procedure, except that the tDCS was pre-set to turn off after 30 s. This procedure produces the expected typical 'tingling' sensation on the scalp and, thus, provides an effective control condition to minimize placebo effects. Regardless of the stimulation type, identical beeping sounds were generated at the beginning and the end of stimulation.

To locate the stimulation regions, individuals' head sizes (see Supplementary Table 2) were measured using a soft measure tape and wax pencil. EZ-EEG[60] (http://clinicalresearcher.org/eeg/) was used to locate the left DLPFC (F3) accurately and efficiently from the nasion-inion, tragus-tragus and circumference lengths. The right PPC (P4) was located at the symmetrical point of left DLPFC (F3), centering at Cz, according to the international 10-20 system[61].

### VWM task

Post-tDCS VWM performance was measured using a continuous orientation-reproduction task (Fig. 1c). In each trial, first a fixation cross was displayed centrally for 1000 ms. Next, an array of two, four or six randomly orientated (0–359°) isosceles triangles that were arranged in a circular manner appeared on the screen for 200 ms. One of the displayed triangles was randomly probed as the target stimulus. After a 1000 ms blank screen, the target stimulus was presented in a random orientation at the same location. Participants were instructed to reproduce the original orientation using the mouse. Reaction time and recall errors (i.e., angular distance between the targeted orientation and reported orientation) were recorded. Note that we did not manipulate the blank interval duration as Wang et al.[1]

did, as they did not observe any effect of tDCS in their short-interval (100 ms, labeled sensory memory) condition.

During each session, participants first completed 30 practice trials (10 practice trials per set size, intermixed) with feedback. For this feedback, the original stimulus array was shown, overlaid by the reproduced angle in green for recall errors smaller than 15 degrees, in orange for errors between 15 and 45 degrees, and in red for errors larger than 45 degrees. Next, participants completed 360 trials without feedback in six blocks (120 trials per set size). Set size was intermixed in the current study, whereas a blocked design was used in the original study, with each block consisting of 60 trials of one set size[1]. The VWM task was executed with Tatool Web[62] (www.tatool-web.com).

### Model fitting

First, we calculated recall errors for each set size and stimulation condition and fitted computational models to these recall errors. Specifically, we compared fits of the Standard Mixture Model[46] (SMM) and Swap Model[63] (SM) to recall errors using the MATLAB MemToolbox[64]. Note that absolute recall errors (from 0 to 180 degrees), which were pre-registered as outcome, were first transformed to directional recall errors (from −180 to 180 degrees), as required by the MemToolbox fitting procedures. Following Wang et al.'s[1] procedure, we computed the Akaike information criterion (AIC) and the Bayesian information criterion (BIC) to indicate relative fits of the models to the data separately for each participant, set size, and stimulation condition. As shown in Table 1, overall, both the AIC and BIC favored the SMM over the SM (60.65% and 87.04%, respectively). The AIC and BIC values of each participant in all conditions are listed in Supplementary Table 3 and Supplementary Table 4, respectively. Then, we used

**Table 1 | Summary of model fits favored the standard mixture model over the swap model**

| Stimulation | Set size | AIC (%) | BIC (%) |
|---|---|---|---|
| Sham | 2 | 83.33 | 95.83 |
| | 4 | 50.00 | 85.42 |
| | 6 | 64.58 | 87.50 |
| PPC | 2 | 85.42 | 97.92 |
| | 4 | 56.25 | 81.25 |
| | 6 | 33.33 | 72.92 |
| DLPFC | 2 | 79.17 | 95.83 |
| | 4 | 45.83 | 85.42 |
| | 6 | 47.92 | 81.25 |
| Overall | – | 60.65 | 87.04 |

*AIC* Akaike information criterion, *BIC* Bayesian information criterion.

the winning model (i.e., SMM) to estimate the capacity and precision parameters using maximum likelihood estimation. The SMM assumes a mixture of two components: a uniform distribution and a circular von Mises distribution. The height of the uniform distribution (*g*) represents random guess responses, which is used to calculate the probability of retrieving the target stimulus (*Pm* = 1−*g*). Capacity (*K*) is the product of *Pm* and the set size (*K* = *Pm\*N*). The standard deviation (SD) of the von Mises distribution represents the precision of the retrieved representation of the target stimulus. A smaller SD is interpreted as higher precision. The precision is denoted by the inverse of the SD ($SD^{-1}$). Following Wang et al.'s[1] procedure, normalized values ($\Delta K$ and $\Delta SD^{-1}$) were used for testing the hypotheses. Normalized values were computed for statistical analyses by subtracting capacity *K* and precision $SD^{-1}$ in the sham condition from those in the active PPC and DLPFC conditions for each set size and participant. All statistical analyses were performed with R Statistical software[65] and R packages rstatix[66], afex[67], effectsize[68], and BayesFactor[69].

### Reporting summary

Further information on research design is available in the Nature Portfolio Reporting Summary linked to this article.

## Results

### No evidence for enhanced VWM capacity and precision by tDCS

Table 2 lists the descriptive statistics for tDCS effects on capacity and precision relative to sham (see Supplementary Table 5 for descriptive statistics of performance after each stimulation). Wang et al.[1] reported selective effects of tDCS relative to sham stimulation over the PPC, but not the DLPFC, on VWM capacity, but not precision. To test whether these effects can be replicated in our study, like Wang et al.[1], we computed the differences in performance between the active stimulation and the sham condition for each participant and set size. Using these difference scores as the dependent variable, we then ran analyses of variance (ANOVAs) with the two within-subjects factors set size (2, 4, 6) and stimulation region (PPC and DLPFC) for each capacity and precision. Where the assumption of sphericity was violated, Greenhouse-Geisser correction was used (denoted by $F_{GG}$). Bayes factors (BFs) using the default prior (Cauchy distribution with *r* = 0.5) and Monte Carlo setting (iterations = 10,000) were calculated to evaluate the strength of evidence for the absence or presence of effects[70,71]. The posterior distribution with a measure of central tendency and credibility interval are summarized in Supplementary Table 6. The sensitivity of the ANOVAs to smaller and larger priors (i.e., Cauchy distribution with *r* = 0.15 and *r* = 1) was assessed, rendering the same patterns of results (Supplementary Table 7). $BF_{10}$ refers to the evidence in favor of the alternative hypothesis that capacity/precision changes relative to sham are not equal to zero, against the null effect that capacity/precision changes are equal to zero.

As shown in Fig. 2, we found no evidence for tDCS-induced changes in capacity or precision. For capacity, in contrast to Wang et al.[1], we found no significant main effects of stimulation region, $F(1, 47) = 0.30$, $p = 0.584$,

**Table 2 | Descriptive statistics of performance changes relative to sham (*N* = 48)**

| Variable | PPC | | | DLPFC | | |
|---|---|---|---|---|---|---|
| | *M* | SD | 95% CI | *M* | SD | 95% CI |
| Capacity difference ($\Delta K$) | | | | | | |
| Set size 2 | −0.02 | 0.09 | [−0.04, 0.01] | −0.02 | 0.11 | [−0.05, 0.01] |
| Set size 4 | −0.01 | 0.44 | [−0.13, 0.12] | −0.06 | 0.49 | [−0.20, 0.08] |
| Set size 6 | −0.14 | 0.72 | [−0.35, 0.07] | 0.00 | 0.88 | [−0.25, 0.26] |
| Precision difference ($\Delta SD^{-1}$) | | | | | | |
| Set size 2 | −0.00 | 0.01 | [−0.01, 0.00] | 0.00 | 0.01 | [−0.00, 0.00] |
| Set size 4 | −0.00 | 0.01 | [−0.00, 0.00] | −0.00 | 0.01 | [−0.00, 0.00] |
| Set size 6 | −0.00 | 0.02 | [−0.01, 0.00] | −0.00 | 0.01 | [−0.01, 0.00] |

Capacity ranges from 0 to the set size; precision ranges from 0 to ∞.
*M* mean, *SD* standard deviation, *CI* confidence interval on the mean value, *PPC* right posterior parietal cortex, *DLPFC* left dorsolateral prefrontal cortex.

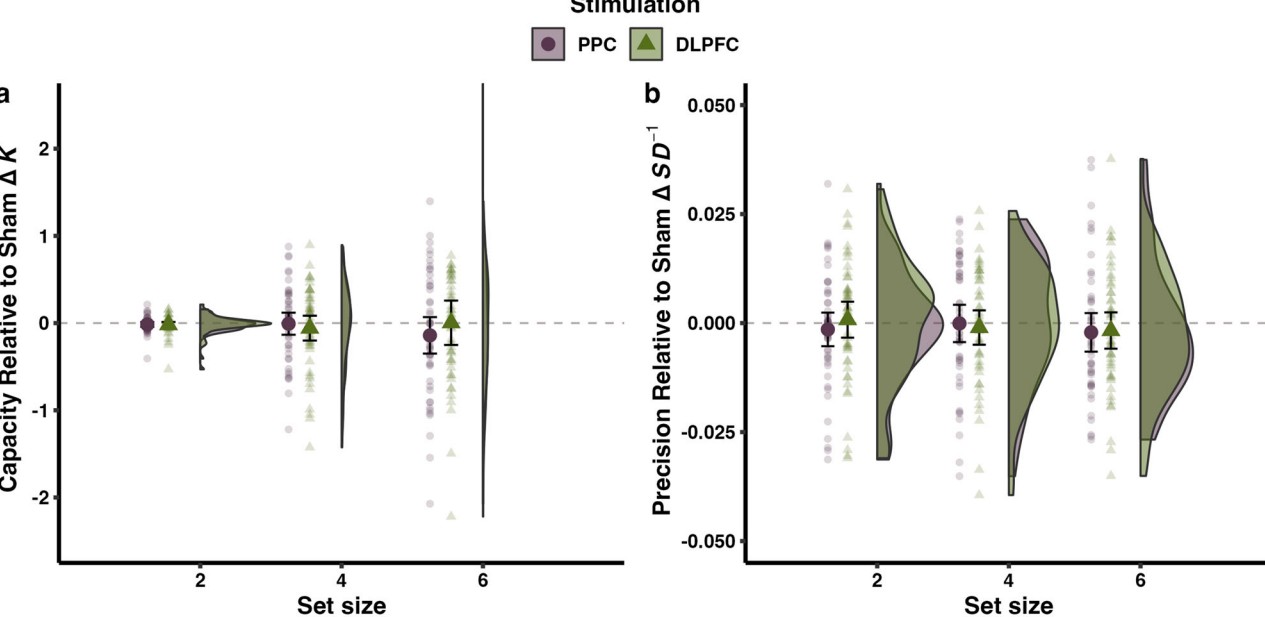

**Fig. 2 | Transcranial direct current stimulation effects on visual working memory capacity and precision relative to sham. a** Changes in capacity relative to sham. **b** Changes in precision relative to sham. Left: opaque symbols indicate group mean values, with the error bars representing 95% confidence intervals. Transparent symbols indicate individual data points. Right: density distributions of the data for both groups.PPC right posterior parietal cortex, DLPFC left dorsolateral prefrontal cortex.

$\eta_G^2 < 0.01$, 95% confidence interval (CI) [0.00, 0.07], $\eta_p^2 = 0.01$, 95% CI [0.00, 0.12], $BF_{10} = 1/6.95 \pm 1.12\%$, and set size, $F_{GG}(1.39, 65.26) = 0.17$, $p = 0.763$, $\eta_G^2 < 0.01$, 95% CI [0.00, 0.02], $\eta_p^2 < 0.01$, 95% CI [0.00, 0.04], $BF_{10} = 1/20.68 \pm 1.94\%$, and no interaction between stimulation region and set size, $F_{GG}(1.18, 55.63) = 1.48$, $p = 0.233$, $\eta_G^2 = 0.01$, 95% CI [0.00, 0.05], $\eta_p^2 = 0.03$, 95% CI [0.00, 0.11], $BF_{10} = 1/6.37 \pm 2.76\%$. Notably, the absence of these effects was supported by substantial to strong Bayesian evidence. If anything, although non-significant, PPC stimulation even induced marginal decreases in capacity relative to sham, opposite to the observed improvements in the original study.

To directly replicate Wang et al.'s[1] analysis on their main findings regarding capacity, we further ran Bonferroni-corrected one-sample t-tests against zero for each region of stimulation and set size (Table 3). Based on the pattern of results from Wang et al.[1] that "enhanced memory capacity via tDCS was specific to PPC (not DLPFC) stimulation" (p. 533), we ran one-sided t-tests for PPC stimulation condition and two-sided t-tests for the DLPFC stimulation condition. Data at set size 2 violated the

**Table 3 | One-sample T-tests for capacity changes relative to sham against zero**

| Stimulation | Set size | Statistical value | *P* value | Effect size | BF | Error (%) |
|---|---|---|---|---|---|---|
| PPC | 2 | 476.00[a] | 0.874 | 0.17 | 1/5.65[b] | <0.01 |
| | 4 | −0.11 | 0.546 | −0.02 | 1/2.65[b] | <0.01 |
| | 6 | −1.35 | 0.909 | −0.20 | 1/20.20[b] | <0.01 |
| DLPFC | 2 | 488.00[a] | 0.424 | 0.12 | 1/2.13[c] | 0.01 |
| | 4 | −0.82 | 0.419 | −0.12 | 1/2.95[c] | 0.01 |
| | 6 | 0.03 | 0.976 | <0.01 | 1/3.92[c] | 0.01 |

df = 47. Bonferroni-corrected threshold of 0.0083.
*PPC* right posterior parietal cortex, *DLPFC* left dorsolateral prefrontal cortex.
[a]Non-Parametric one-sample Wilcoxon signed rank test.
[b]$BF_{10}^{+0}$: in favor of the alternative hypothesis that capacity changes relative to sham are greater than zero, against the null effect that capacity changes equal to zero.
[c]$BF_{10}$: in favor of the alternative hypothesis that capacity changes relative to sham not equal to zero, against the null effect that capacity changes equal to zero.

assumptions, therefore, the equivalent non-parametric one-sample Wilcoxon signed rank test was run for the condition of set size 2. BFs were calculated using default Monte Carlo setting (iterations = 10,000) and informative priors based on the reported significant effect sizes in Wang et al.[1]. Again, based on the pattern of results from the original study, we used Bayes factors ($BF_{10}^{+0}$ and $BF_{10}$) to quantify the strength of evidence for PPC and DLPFC stimulation, respectively. $BF_{10}^{+0}$ refers to the evidence in favor of the alternative hypothesis that capacity changes relative to sham are greater than zero, against the null effect that capacity changes equal to zero. For the PPC condition, we used the reported effect size ($d = 1.028$) as the informative prior. For the DLPFC condition, only the range of effect sizes ($ds = 0.078$–$0.409$) was reported in the original study. Thus, we used the biggest effect size value as the informative prior (Cauchy distribution with $r = 0.409$). The posterior distributions with a measure of central tendency and credibility interval are summarized in Supplementary Table 8. The sensitivity of these follow-up analyses to conventional small, medium, and large priors (i.e., Cauchy distribution with $r = 0.2$, $r = 0.5$, and $r = 0.8$) was assessed, yielding the same patterns of results (Supplementary Table 9).

We observed neither PPC stimulation nor DLPFC stimulation effects compared to sham at any set size. Critically, in contrast to Wang et al.'s[1] main finding that PPC stimulation increased relative capacity changes compared to zero at set size 6 with a large effect size (Cohen's $d = 1.028$), this effect was absent in our data, $t(47) = -1.35$, $p = 0.909$, Cohen's $d = -0.20$, 95% CI [-0.43, ∞], which was supported by strong Bayesian evidence, $BF_{10}^{+0} = 1/20.20 \pm 0.00\%$. Again, if anything, PPC stimulation tended toward *decreasing* capacity at set size 6. Furthermore, Wang et al.[1] reported that the increase in capacity induced by PPC stimulation was significantly higher than that by DLPFC stimulation at set size 6, with a medium to large effect size (Cohen's $d = 0.711$). Note that this test was not explicitly mentioned in the analysis plan of the pre-registration but is necessary to fully test Hypothesis 1. Different to the original study, we found no credible evidence of a significant difference in relative capacity changes between the effects of the two stimulation sites at set size 6, $t(47) = -1.03$, $p = 0.846$, Cohen's $d = -0.15$, 95% CI [-0.39, ∞]. The absence of this difference was supported by strong Bayesian evidence, $BF_{10}^{+0} = 1/12.14 \pm 0.00\%$.

Regarding tDCS effects on precision, we did not observe significant effects for any stimulation region or set size, consistent with the original results and our Hypothesis 2. A two-way repeated measures ANOVA showed no significant main effects of stimulation region, $F(1,47) = 0.15$, $p = 0.700$, $\eta_G^2 < 0.01$, 95% CI [0.00, 0.06], $\eta_p^2 < 0.01$, 95% CI [0.00, 0.10], $BF_{10} = 1/7.38 \pm 0.84\%$, or set size, $F_{GG}(1.84, 86.36) = 0.33$, $p = 0.700$, $\eta_G^2 < 0.01$, 95% CI [0.00, 0.03], $\eta_p^2 = 0.01$, 95% CI [0.00, 0.06], $BF_{10} = 1/17.14 \pm 1.25\%$. There was also no significant interaction effect, $F_{GG}(1.96, 92.15) = 0.85$, $p = 0.430$, $\eta_G^2 < 0.01$, 95% CI [0.00, 0.03], $\eta_p^2 = 0.02$, 95% CI [0.00, 0.09], $BF_{10} = 1/10.58 \pm 1.80\%$. The absence of these effects was supported by substantial to strong Bayesian evidence.

## Summary

Like Wang et al.[1], we observed no effects of tDCS on VWM capacity and precision induced by DLPFC stimulation. However, in contrast to the original study, PPC stimulation did not significantly enhance capacity selectively at set size 6, and also not at any other set size. The absence of tDCS effects was largely supported by substantial to strong Bayesian evidence.

## Discussion

This preregistered study aimed to replicate the benefits of non-invasive brain stimulation on VWM that were recently reported by Wang et al.[1]. Wang et al.[1] found that tDCS over the PPC, but not the DLPFC, selectively improved capacity, but not precision, when VWM load was high (set size 6). While this conceptual replication accounted for methodological issues from the original study, we found no credible evidence of such selective effects. Stimulation over the right PPC improved neither capacity nor precision of representations in VWM performance. In contrast, if anything, our results indicated that when VWM load is high, capacity slightly, although not significantly, decreased after right PPC stimulation compared to the sham condition. Therefore, our Hypothesis 1 (improvements of capacity) was rejected, while Hypothesis 2 (no improvements in precision) was confirmed.

With the present study being a conceptual, and not a direct, replication of Wang et al.'s[1] study, there are few notable methodological differences between the two studies, which are summarized in Table 4. A particularly striking difference is the sample size, which was about 2.5 times bigger in the present than in the original study. The lack of a PPC-tDCS effect on VWM in the present study, which was supported by unambiguously strong Bayesian evidence, suggests that the original findings may have been false-positive results. The two samples were comparable in age and their baseline VWM performance. This is important because

differences in baseline VWM capacity may contribute to the differences in results between the replication study and the original study. Previous studies showed that only low-performing participants benefited from anodal PPC stimulation[8,12]. However, based on the descriptive data available from the original study (see Fig. 2 in Wang et al.[1]), baseline performance was comparable and, if anything, slightly lower in the present study.

As discussed above, we deliberately modified the design of the administered task to address methodological issues that we identified in the original study. Critically, these modifications should either not affect or increase the likelihood of observing PPC-tDCS effects. First, although different stimuli (triangles vs bars) were used, both the current study and the original study tested the memory of orientation information. Both triangles and bars are basic two-dimensional shapes that people are familiar with. Therefore, the slight variation in the stimuli's shape *per se* is unlikely to cause the absence of PPC-tDCS VWM benefits. In fact, if tDCS benefits were robust and meaningful, they ideally should be generalizable to different stimuli and even paradigms. We chose triangles over bars as stimuli to increase the difficulty level of the VWM task, as the triangles allow for using the full space of possible orientations. Previous studies have indicated greater tDCS benefits for more challenging tasks, possibly due to more room for performance improvement[10,48,49]. Hence, if anything, a more difficult VWM task design is more likely to lead to greater tDCS benefits. Yet, we did not replicate the benefits of PPC stimulation. Second, another key difference in the VWM task design is that set size conditions were intermixed in the current study but presented in blocks in the original study. Being able to focus on only one set size condition per block, participants from the original study may have been more likely to develop effective encoding strategies, especially at higher set sizes[72].

Our findings are consistent with recent studies that focused on other stimulation sites and used different paradigms, suggesting that the lack of an effect in our study is not specific to the present montage or paradigm. For instance, the absence of benefits of anodal PPC-tDCS in the present study is consistent with other recent research[31,32] that used other types of montage (left PPC) and/or VWM paradigms (change-detection). Like Dumont et al.[31], our findings were largely supported by substantial to strong Bayesian evidence, challenging the previously reported positive effects of anodal PPC-tDCS on VWM capacity[1]. Our results are also consistent with the results from Nikolin et al.[33] that anodal DLPFC stimulation does not alter VWM performance, even using a different montage (i.e., cathode on the contralateral DLPFC) and VWM paradigm (n-back). Nikolin et al.[33] further systematically tested different tDCS dosages, resulting in consistent null effects. Our findings add critically to the existing literature by demonstrating

## Table 4 | Summary of key differences

| Study characteristic | Current study | Wang et al.[1] |
|---|---|---|
| Participants | | |
| Sample size for analysis | 48 | 18 |
| Demographic characteristics | 31 female participants, 17 male participants; Age 22.65 ± 4.34 years old, range 18–33 years old; Residence in UK | 14 female participants; Age, 22.9 ± 1.94 years old; Residence in China |
| Baseline performance (sham) | Supplementary Table 5 | Fig. 2 (p. 532) |
| VWM Task | | |
| Stimulus | Triangles | Bars |
| Orientation space | Randomly from 0° to 359°; at least 1° difference between one another | Randomly from 10° to 170°, at least 10° difference between one another |
| Number of trials | 120 trials per set size; intermixed | 60 trials per set size; blocked |
| Procedure | Short tDCS questionnaires between stimulation and VWM task (<5 mins) | |
| tDCS device | | |
| Apparatus | TCT research tDCS 1ch | Eldith NeuroConn |
| Ramp up/down | 30 s/2 s | 20 s/20 s |

that it is unlikely that anodal DLPFC-tDCS produces improvements in VWM performance in healthy participants.

A simple possible explanation for the difficulties to replicate tDCS effects on cognitive functioning is their lack of a consistent physiological basis. For example, the finding that anodal stimulation increases cortical excitability and cathodal stimulation decreases cortical excitability has often been replicated with unilateral, low-intensity (1 mA) stimulation. However, these classic polarity-specific effects did not extend to higher-intensity stimulation at 2 mA[73] or bilateral stimulation[74].

Overall, our findings highlight the importance of replication studies investigating robust tDCS effects. Wang et al.'s[1] findings would have far-reaching theoretical and practical implications for the scientific understanding of both tDCS effects and VWM processes. Our findings mirror the replication crisis that replication effect sizes are typically only a quarter or halve of the magnitude of original effects[75,76]. Our replication attempt—which disconfirmed tDCS-induced increases in VWM capacity by stimulating right PPC—can serve as a starting point for more replications to further test the veracity of such tDCS effects[77]. Similarly, Robison et al.[32] recently failed to conceptually replicate the positive tDCS effects over PPC and DLPFC that were reported by Li et al.[30], using a design similar but not identical to the original one. Altogether, these two examples of conceptual replication attempts are likely only the tip of the iceberg of a lack of replicability and generalization in tDCS research. Importantly though, any single replication study does not rule out that tDCS may benefit cognitive performance in general[78,79]. Therefore, more replications using the same tDCS setups are needed to advance this promising area of research.

## Limitations

There are a few minor differences in procedural details and the tDCS device settings that need noting as possible limitations, even though they are unlikely to explain the lack of PPC-tDCS benefits relative to the original study. First, unlike the original study where participants immediately completed the VWM task after stimulation, participants in the current study completed short questionnaires (well under 5 min) after stimulation. This procedure is designed to measure the safety of tDCS and exclude possible adverse tDCS effects impacting the observed tDCS effects, which is a requirement for ethical approval at our institution. However, effects of tDCS of more than 10 mins typically last longer than an hour[80–82]. Hence, the short interruption by the questionnaires is highly unlikely to impact the post-stimulation effects on VWM performance. Second, the tDCS devices used in the present and the original study differed in their apparatus. Both types of tDCS devices were widely used in previously studies (TCT[52,83,84]; NeuroConn[85–87]). However, the devices differ in their ramp-up (30 s vs 20 s) and ramp-down (2 s vs 20 s) time, which are used to mimic cutaneous sensations that are associated with changing current, and thus to provide good control of condition blindness[88,89]. There is no consensus in the literature regarding ramp-up and ramp-down settings and our questionnaire findings confirmed a good level of condition blindness (see Supplementary Notes 1).

## Conclusions

We did not observe any benefits of single-session, anodal parietal or prefrontal tDCS on VWM capacity and precision. In particular, we found no evidence for the selective, large effect of parietal tDCS in increasing VWM capacity at a big set size that was reported by Wang et al.[1]. Indeed, the empirical evidence from our study consistently favored the absence of any cognitive benefits after tDCS regardless of stimulation site and task difficulty. Considering the complexity of tDCS parameters and setups, our null findings highlight the critical importance of conducting replications for building a robust and informative body of evidence on the effectiveness of non-invasive brain stimulation on cognitive performance.

## Data availability
The data that support the findings of this study and supplementary information are openly available on OSF under this link: https://osf.io/92k4w/[90].

## Code availability
All code for running the experiment, data cleaning, and analysis associated with the current submission is available on OSF under this link: https://osf.io/92k4w/[90].

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

## Acknowledgements

We thank Tyler Mari who piloted this study as part of his MSc thesis. We also acknowledge the support from the Economic and Social Research Council (ESRC) to C.C.v.B. (ES/V013610/1) and the Open Research Award from the University of Sheffield. The funders had no role in study design, data collection and analysis, and decision to publish or preparation of the manuscript.

## Author contributions

Shuangke Jiang: conceptualization, data curation, formal analysis, investigation, methodology, project administration, resources, software, validation, visualization, writing—original draft, and writing—review & editing. Myles Jones: conceptualization, funding acquisition, methodology, resources, supervision, and writing—review & editing. Claudia C. von Bastian: conceptualization, funding acquisition, methodology, resources, software, supervision, and writing—review & editing.

## Competing interests

The authors declare no competing interests. Claudia C. von Bastian is an Editorial Board Member for Communications Psychology, but was not involved in the editorial review of, nor the decision to publish this article.
