## [Peer Review File · Communications Psychology]

14th Sep 23

Dear Ms Jiang,

Thank you for your patience during the peer-review process. Your manuscript titled "Does transcranial direct current stimulation enhance visual working memory? A replication study" has now been seen by 3 reviewers, and I include their comments at the end of this message. They find your work of interest, but raised some important points. We are interested in the possibility of publishing your study in *Communications Psychology*, but would like to consider your responses to these concerns and assess a revised manuscript before we make a final decision on publication.

We therefore invite you to revise and resubmit your manuscript, along with a point-by-point response to the reviewers. Please highlight all changes in the manuscript text file.

The reviewers find your replication study well motivated and technically sound. They provide some suggestions to strengthen your rationale and discussion, which we hope you will find useful.

We have also solicited feedback from Drs Sisi Wang and Yixuan Ku in the form of signed comments. These are attached separately.

Please note that signed comments do not function as reviews. The authors of signed comments are asked to comment solely on methodological aspects of your work, as well as on the extent to which their work is accurately represented in your manuscript. Decisions on whether to reject a manuscript or invite a revision are based on the feedback of our independent reviewers. However, if a decision to invite a revision is made, signed comments inform our requests for necessary revisions. (For full details on our Signed Comments policy, please see <https://www.nature.com/commspsychol/referees/signed-comments>)

In response to the signed comments by Drs Wang and Ku, we ask you to mention in the Abstract that the replication isn't exact but contains a slight variation of the paradigm and setup. Further, please discuss these differences and how they might potentially account for differences in the result (or not) in the Discussion section.

Please use the following link to submit your revised manuscript, point-by-point response to the referees' comments (which should be in a separate document to any cover letter) and the completed checklist:

[link redacted]

We hope to receive your revised paper within 8 weeks; please let us know if you aren't able to submit it within this time so that we can discuss how best to proceed. If we don't hear from you, and

the revision process takes significantly longer, we may close your file. In this event, we will still be happy to reconsider your paper at a later date, provided it still presents a significant contribution to the literature at that stage.

Please do not hesitate to contact me if you have any questions or would like to discuss these revisions further. We look forward to seeing the revised manuscript and thank you for the opportunity to review your work.

Best regards,

Antonia Eisenkoeck

Antonia Eisenkoeck
Senior Editor
Communications Psychology

EDITORIAL POLICIES AND FORMATTING

Editorial Policy: Policy requirements (Download the link to your computer as a PDF.)

Furthermore, please align your manuscript with our format requirements, which are summarized on the following checklist:

Communications Psychology formatting checklist

and also in our style and formatting guide Communications Psychology formatting guide .

* **CODE AVAILABILITY:** All Communications Psychology manuscripts must include a section titled "Code Availability" at the end of the methods section. In the event of publication, we require that the custom analysis code supporting your conclusions is made available in a publicly accessible repository; at publication, we ask you to choose a repository that provides a DOI for the code; the link to the repository and the DOI will need to be included in the Code Availability statement.

Publication as Supplementary Information will not suffice. We ask you to prepare code at this stage, to avoid delays later on in the process.

* DATA AVAILABILITY:

All Communications Psychology manuscripts must include a section titled "Data Availability" at the end of the Methods section or main text (if no Methods). More information on this policy, is available at <http://www.nature.com/authors/policies/data/data-availability-statements-data-citations.pdf>.

At a minimum the Data availability statement must explain how the data can be obtained and whether there are any restrictions on data sharing. Communications Psychology strongly endorses open sharing of data. If you do make your data openly available, please include in the statement:

We recommend submitting the data to discipline-specific, community-recognized repositories, where possible and a list of recommended repositories is provided at <http://www.nature.com/sdata/policies/repositories>.

If a community resource is unavailable, data can be submitted to generalist repositories such as figshare or Dryad Digital Repository. Please provide a unique identifier for the data (for example a DOI or a permanent URL) in the data availability statement, if possible. If the repository does not provide identifiers, we encourage authors to supply the search terms that will return the data. For data that have been obtained from publicly available sources, please provide a URL and the specific data product name in the data availability statement. Data with a DOI should be further cited in the methods reference section.

REVIEWERS' EXPERTISE:

Reviewer #1: tDCS

Reviewer #2: tDCS

Reviewer #3: visual working memory

REVIEWERS' COMMENTS:

Reviewer #1 (Remarks to the Author):

First of all, my apologies for such a slow response on a relatively simple paper. The issue of the apparently wide-ranging effects of TDCS is an important one to address, particularly because claims are made in therapeutic areas of particular consequence and vulnerability (autism, depression, ADHD, dementia, stroke etc).

One of the features of the literature, as correctly pointed out in this submission, is the frequent use

of small group sizes and reporting of weak, but statistically positive effects.

The nature of the field (by which I really mean the fact that there isn't large industrial money available) precludes large clinical trials of the kinds one might see in studies of drugs which make similar claims. It's therefore particularly important that when moving from the experimental to the clinical, preregistered studies used as an opportunity of testing clinical claims in the absence of larger scale multicenter trials.

This article presents affair and straightforward like for like replication challenge to the claims concerning visual working memory. Given the low signal to noise ratio in this literature, it's difficult to see any objection to the experiment as conceived, presented, and interpreted in this submission. In some areas of research pre registration and replication/non replication sometimes offer little added value, but where consequential clinical claims are made it is an important test of delivery.

I have only one suggestion to make regarding the context of the discussion. It is indeed useful to have claims and counter claims about the effects of stimulation. But there is a now decade-long contention that the claims of TDCS effects on cognitive functioning are without physiological foundation. A study of the effects of TDCS intensity and polarity on the motor cortex concluded that the anodal-cathodal polarity assumptions in these studies are reversed at 2mA (Batsikadze et al., 2013). A later study showed that this absence of 2mA effects in cognitive experiments (Parkin et al., 2013). It may be worth noting that one of the reasons that the enhancement in VWM was observed in the current study is that the presumed polarity of currents does not exist at 2mA.

Batsikadze, G., Moliadze, V., Paulus, W., Kuo, M.-F., Nitsche, M.A., 2013. Partially non-linear stimulation intensity-dependent effects of direct current stimulation on motor cortex excitability in humans. *J. Physiology* 591 (Pt7), 1987–2000. <http://dx.doi.org/10.1113/jphysiol.2012.249730>.

Parkin, B., Bhandari, M., Glen, J.C., Walsh, V. 2019. The physiological effects of transcranial electrical stimulation do not apply to parameters commonly used in studies of cognitive neuromodulation. *Neuropsychologia*, 128, 332-339,

Reviewer #2 (Remarks to the Author):

In the present study, the authors investigated the effectiveness of anodal transcranial direct current stimulation (tDCS) over the left dorsolateral prefrontal cortex (DLPFC) and the right posterior parietal cortex (PPC) to improve visual working memory (VWM). The study was, in large part, an effort to replicate a recent study by Wang et al. (2019) showing large improvements to VWM at large set sizes, specifically with tDCS to the right PPC. In an attempt to replicate this effect, the authors recruited a larger (N = 48) sample size, extended the number of trials, and slightly altering the stimuli to allow for 360-degree estimation of orientation. In a fully within-subjects design, they delivered active tDCS over the right PPC, active tDCS over the left DLPFC, or sham stimulation. Stimulation conditions were delivered in separate sessions. In all sessions, there was a 15-minute period of stimulation/sham, then a 30-minute VWM task with set sizes 2, 4, and 6. The task was a continuous-report orientation task in which the participant needed to reproduce the presented angle of an isosceles triangle. Performance was assessed by fitting the responses to a standard two-parameter mixture model. The authors observed no evidence for an improvement to VWM, and in

fact there was substantial evidence in favor of the null hypothesis of no improvement with Bayesian analyses.

Overall, my impression of the manuscript was very positive. In fact, I have no substantive recommendations for revision. While I was reading it, every time I had a question, it was subsequently answered somewhere in the manuscript. The strengths include a preregistration of the hypothesis, a well-considered experimental design and sample size, a thorough analysis of the data, and a comprehensive review of the extant literature. I very rarely do this, but I recommend this paper be accepted without any minor or major revisions.

Reviewer #3 (Remarks to the Author):

This study aims to replicate Wang et al.'s (2019) study that used tDCS over PPC to improve visual working memory (VWM) capacity. I think the reasoning for a replication study is well justified by good theoretical and statistical reasoning. The study design also corresponds to the points raised (e.g., sample size, counterbalance sessions, SOA...etc). I do think the analysis and discussion can be improved to provide more information. As of now, the study looks like a simple "what" study (i.e., there is no tDCS effect) when it can actually offer more "why". My comments are below:

Need to explain something, like Hsu Tseng EEG observation

1. Like the authors have pointed out, Wang et al.'s (2019) original claim contradicts many previous findings on DLPFC tDCS (Andrews et al., 2011; Fregni et al., 2005; Ohn et al., 2008; Zaehle et al., 2011). But this is simply stated, and not explained. I wonder if the authors have any insights into the methodological differences that may have given rise to these inconsistencies.
2. Page 8, Line 160: this is an interesting point as remembering one end the defeat the purpose of precision (becomes a location VWM task), therefore the null result in precision from Wang et al. may also be due to poor task design. Of course, this would predict that, once fixed, the positive effect should emerge, which is not the case in the present study. But nevertheless, I think this is fair game at least in the Introduction, and it would benefit the readers to see that task design really matters (perhaps even more so in tDCS studies).
3. Continuing from above, I can understand why the authors choose not to use the rotating bar task. But, I'm not sure if the triangle stimuli here solves the location-VSTM problem mentioned above, as the participants can simply remember where the tip is? Wouldn't that have the same problem as Wang et al's study? It seems that a color wheel precision task may have been more appropriate here?
4. I think more attempts on speculating on the "why" is welcomed in the Discussion. Do the authors think it's one (or many) of the methodological issues that caused a non-replication, or simply a false positive due to low statistical power from the original study?
5. In the Discussion, the authors also challenges other PPC tDCS studies such as Tseng et al (2012) and Hsu et al. (2014), but it seems that several aspects of the current study are quite different from

those studies (e.g., how to explain their EEG findings; low vs. high performing individual differences in tDCS receptivity; color location task) need to be accounted for?

Signed Comments for “Does transcranial direct current stimulation enhance visual working memory? A replication study”

Yixuan Ku¹, Sisi Wang²

1. Department of Psychology, Sun Yat-sen University, Guangzhou, China

2. Department of Experimental and Applied Psychology, Vrije Universiteit Amsterdam, The Netherlands

It appears that Jiang et al. conducted a study that they referred to as a "replication study" of our original research, but there are several key differences between their “replication” study and our original study. These differences are as follows:

1. Distinct Task Paradigm: Jiang et al. used a different task paradigm compared to our original study. They employed a spatial working memory task with triangles, whereas our original study focused on a different task involving the orientation of square bars. In the case of square bars, which possess equal lengths on both the top and bottom, participants could easily disregard the shape of the stimuli and concentrate solely on retaining the orientation of the bar. In contrast, when dealing with triangle stimuli, the cognitive processing demands are heightened as participants must simultaneously process both shape and orientation information. This disparity in cognitive processing requirements between the two task paradigms is a significant factor to consider. The use of different stimuli and cognitive processes involved in these tasks may lead to different outcomes regarding the effects of tDCS on the posterior parietal cortex (PPC).

2. Different Task Design: Our original study used a block design, where each block consisted of 60 trials with a specific memory set size. In contrast, Jiang et al. used a mixed design, where different set sizes were mixed within each block. This could potentially lead to shifts in mental workload and among different set sizes, complicating the interpretation of the tDCS effects. The lack of original working memory performance data in Jiang et al.'s study also complicates the interpretation. The data in the sham condition should be illustrated to assess how well participants performed the tasks.

3. Different Task Procedure: There was a significant difference in the timing of the task relative to tDCS stimulation. In our original study, participants started the memory task immediately after tDCS stimulation ended to maximize the stimulation effect. In contrast, Jiang et al. had participants complete questionnaires after stimulation, which introduced a delay and different cognitive processes that may impact the tDCS effect on PPC.

4. Difference in Stimulating Apparatus: Jiang et al. used a different tDCS stimulator, namely "TCT Research tDCS 1ch device," while we used the neuroConn DC-stimulator, which is widely recognized and used in many studies. Differences in tDCS equipment can affect the application and effectiveness of tDCS and may contribute to variations in study outcomes.

5. Reference to Other Studies: We successfully replicated our initial findings and also observed the influence of an additional strategy in a follow-up study (Wang et al., 2020). Other studies have reported similar stimulating effects on PPC in different domains, such as spatial (Zhu et al., 2022) and color (Pailian & Alvarez, 2019)

working memory tasks. This suggests that the use of tDCS to investigate the role of PPC in working memory storage is not unique to our study.

In summary, while Jiang et al. referred to their study as a "replication study", there are significant differences in key aspects of the experimental design, task paradigm, stimulating apparatus, and procedure compared to our original study. These differences could potentially account for variations in the results and interpretations between the two studies. It's important to consider these differences when assessing the reproducibility and generalizability of research findings. Furthermore, it is necessary for the authors to present their data from the sham condition, enabling others to evaluate participants' task performance and its variations.

Reference:

- Pailian, H. & Alvarez, G. A. (2019). Probing the Neurocognitive Architecture of Visual Working Memory by Enhancing Storage vs. Manipulation Abilities. *Journal of Vision*, 19(10), 247. <https://doi.org/10.1167/19.10.247>
- Wang, S., Itthipuripat, S. & Ku, Y. (2020). Encoding strategy mediates the effect of electrical stimulation over posterior parietal cortex on visual short-term memory. *Cortex*, 128, 203–217. <https://doi.org/10.1016/j.cortex.2020.03.005>
- Zhu, R., Luo, Y., Wang, Z. & You, X. (2022). Within-session repeated transcranial direct current stimulation of the posterior parietal cortex enhances spatial working memory. *Cognitive Neuroscience*, 13(1), 26–37. <https://doi.org/10.1080/17588928.2021.1877648>

Reviewer #1 (Remarks to the Author):

First of all, my apologies for such a slow response on a relatively simple paper. The issue of the apparently wide-ranging effects of TDCS is an important one to address, particularly because claims are made in therapeutic areas of particular consequence and vulnerability (autism, depression, ADHD, dementia, stroke etc).

One of the features of the literature, as correctly pointed out in this submission, is the frequent use of small group sizes and reporting of weak, but statistically positive effects.

The nature of the field (by which I really mean the fact that there isn't large industrial money available) precludes large clinical trials of the kinds one might see in studies of drugs which make similar claims. It's therefore particularly important that when moving from the experimental to the clinical, preregistered studies used as an opportunity of testing clinical claims in the absence of larger scale multicenter trials.

This article presents affair and straightforward like for like replication challenge to the claims concerning visual working memory. Given the low signal to noise ratio in this literature, it's difficult to see any objection to the experiment as conceived, presented, and interpreted in this submission. In some areas of research pre registration and replication/non replication sometimes offer little added value, but where consequential clinical claims are made it is an important test of delivery.

I have only one suggestion to make regarding the context of the discussion. It is indeed useful to have claims and counter claims about the effects of stimulation. But there is a now decade-long contention that the claims of TDCS effects on cognitive functioning are without physiological foundation. A study of the effects of TDCS intensity and polarity on the motor cortex concluded that the anodal-cathodal polarity assumptions in these studies are reversed at 2mA (Batsikadze et al., 2013). A later study showed that this absence of 2mA effects in cognitive experiments (Parkin et al., 2013). It may be worth noting that one of the reasons that the enhancement in VWM was observed in the current study is that the presumed polarity of currents does not exist at 2mA.

Batsikadze, G., Moliadze, V., Paulus, W., Kuo, M.-F., Nitsche, M.A., 2013. Partially non-linear stimulation intensity-dependent effects of direct current stimulation on motor cortex excitability in humans. *J. Physiology* 591 (Pt7), 1987–2000. <http://dx.doi.org/10.1113/jphysiol.2012.249730>.

Parkin, B., Bhandari, M., Glen, J.C., Walsh, V. 2019. The physiological effects of transcranial electrical stimulation do not apply to parameters commonly used in studies of cognitive neuromodulation. *Neuropsychologia*, 128, 332-339.

Reply:	We thank the reviewer for their insightful and positive comments and this suggestion. We agree with the concerns regarding polarity assumptions which we have also briefly mentioned in the introduction. Specifically, we pointed out “...whereas excitatory effects of anodal stimulation are largely robust, inhibitory effects of cathodal stimulations are less consistent...” (p. 4, line 70 ff.). The recommended references are good additions to strengthen this point and provide a possible explanation for the non-replicable effects of tDCS. Therefore, we added the following content in the discussion: “A simple possible explanation for the difficulties to replicate tDCS effects on cognitive functioning is their lack of a consistent physiological basis. For example, the finding that anodal stimulation increases cortical excitability and cathodal stimulation decreases cortical excitability has often been replicated with unilateral, low-intensity (1 mA) stimulation. However, these classic polarity-specific effects did not extend to higher-intensity stimulation at 2 mA (Batsikadze et al., 2013) or bilateral stimulation (Parkin et al., 2019).”(p. 19, line 367 ff.)
---------------	---

Reviewer #2 (Remarks to the Author):

In the present study, the authors investigated the effectiveness of anodal transcranial direct current stimulation (tDCS) over the left dorsolateral prefrontal cortex (DLPFC) and the right posterior parietal cortex (PPC) to improve visual working memory (VWM). The study was, in large part, an effort to replicate a recent study by Wang et al. (2019) showing large improvements to VWM at large set sizes, specifically with tDCS to the right PPC. In an attempt to replicate this effect, the authors recruited a larger (N = 48) sample size, extended the number of trials, and slightly altering the stimuli to allow for 360-degree estimation of orientation. In a fully within-subjects design, they delivered active tDCS over the right PPC, active tDCS over the left DLPFC, or sham stimulation. Stimulation conditions were delivered in separate sessions. In all sessions, there was a 15-minute period of stimulation/sham, then a 30-minute VWM task with set sizes 2, 4, and 6. The task was a continuous-report orientation task in which the participant needed to reproduce the presented angle of an isosceles triangle. Performance was assessed by fitting the responses to a standard two-parameter mixture model. The authors observed no evidence for an improvement to VWM, and in fact there was substantial evidence in favor of the null hypothesis of no improvement with Bayesian analyses.

Overall, my impression of the manuscript was very positive. In fact, I have no substantive recommendations for revision. While I was reading it, every time I had a question, it was subsequently answered somewhere in the manuscript. The strengths include a preregistration of the hypothesis, a well-considered experimental design and sample size, a thorough analysis of the data, and a comprehensive review of the extant literature. I very rarely do this, but I recommend this paper be accepted without any minor or major revisions.

Reply:	We thank the reviewer for their kind comments and enthusiastic appraisal of our study.
---------------	--

Reviewer #3 (Remarks to the Author):

This study aims to replicate Wang et al.'s (2019) study that used tDCS over PPC to improve visual working memory (VWM) capacity. I think the reasoning for a replication study is well justified by good theoretical and statistical reasoning. The study design also corresponds to the points raised (e.g., sample size, counterbalance sessions, SOA...etc). I do think the analysis and discussion can be improved to provide more information. As of now, the study looks like a simple "what" study (i.e., there is no tDCS effect) when it can actually offer more "why". My comments are below:

Reply:	We thank the reviewer for their valuable comments which have hugely helped us to improve the manuscript.
---------------	--

Need to explain something, like Hsu Tseng EEG observation

1. Like the authors have pointed out, Wang et al.'s (2019) original claim contradicts many previous findings on DLPFC tDCS (Andrews et al., 2011; Fregni et al., 2005; Ohn et al., 2008; Zaehle et al., 2011). But this is simply stated, and not explained. I wonder if the authors have any insights into the methodological differences that may have given rise to these inconsistencies.

Reply:	We thank the reviewer for this suggestion which we totally agree with. Thus, we have revised this part and added the further discussion on these inconsistencies in terms of methodological differences: “First, the findings from this study falsified the role of the anodal stimulation at 2 mA on DLPFC in improving VWM, thereby contradicting previous studies in which a weaker current (1 mA) and different WM paradigms (digits forward span and 1–3-back) were applied (Andrews et al., 2011; Fregni et al., 2005; Ohn et al., 2008; Zaehle et al., 2011). Indeed, recent studies reported an overall absence of anodal DLPFC-tDCS effects on enhancing WM performance regardless of current intensity (Nikolin et al., 2018; Papazova et al., 2020). Furthermore, one consistent finding regarding differences in task type across several studies is that individuals benefit from tDCS when WM tasks are more demanding (Jones & Berryhill, 2012; Papazova et al., 2020; Wu et al., 2014).” (p. 7, line 124 ff.)
---------------	--

2. Page 8, Line 160: this is an interesting point as remembering one end the defeat the purpose of precision (becomes a location VWM task), therefore the null result in precision from Wang et al. may also be due to poor task design. Of course, this would predict that, once fixed, the positive effect should emerge, which is not the case in the present study. But nevertheless, I think this is fair game at least in the Introduction, and it would benefit the readers to see that task design really matters (perhaps even more so in tDCS studies).

Reply:	We agree that task design is important in tDCS studies. Thus we have added more in-depth discussion on the key differences in VWM task design between the current study and the original study to address the reviewer’s comment. “As discussed above, we deliberately modified the design of the administered task to address methodological issues that we identified in the original study. Critically, these modifications should either not affect or increase the likelihood of observing PPC-tDCS effects. First, although different stimuli (triangles vs bars) were used, both the current study and original study tested the memory of orientation information. Both triangles and bars are basic two-dimensional shapes that people are familiar with. Therefore, the slight variation in the stimuli’s shape per se is unlikely to cause the absence of PPC-tDCS VWM benefits. In fact, if tDCS benefits were robust and meaningful, they ideally should be generalisable to different stimuli and even paradigms. We chose triangles over bars as stimuli to increase the difficulty level of the VWM task, as the triangles allow for using the full space of possible orientations. Previous studies have indicated greater tDCS benefits for more challenging tasks, possibly due to more room for performance improvement (Jones & Berryhill, 2012; Papazova et al., 2020; Wu et al., 2014). Hence, if anything, a more difficult VWM task design is more likely to lead to greater tDCS benefits. Yet, we did not replicate the benefits of PPC stimulation. Second, another key difference in the VWM task design is that set size conditions were intermixed in the current study but presented in blocks in the original study. Being able to focus on only one set size condition per block, participants from the original study may have been more likely to develop effective encoding strategies, especially at higher set sizes (see Wang et al., 2020). ” (p. 17, line 315 ff.)
---------------	---

3. Continuing from above, I can understand why the authors choose not to use the rotating bar task. But, I’m not sure if the triangle stimuli here solves the location-VSTM problem mentioned above, as the participants can simply remember where the tip is? Wouldn’t that have the same problem as Wang et al’s study? It seems that a color wheel precision task may have been more appropriate here?

Reply:	We deliberately chose to use an orientation task to avoid that any divergent results were simply due to using stimuli from different domains. Indeed, there is evidence that orientation information is processed and stored differently than colour information (see Ricker et al., 2023). Thus, we instead aimed at improving the paradigm used in the original study. While we agree that participants in our study could also still simply remember where the tip was, the use of the full orientation space of 360 possible answers considerably reduces the chance level. To clarify this also in the manuscript, we have added the following: “Finally, Wang et al. (2019) used rotated bars as stimuli. The unique angles of their stimuli effectively ranged only from “10° to 170°” (p. 529), leaving room for developing task-specific strategies. For example, participants may have realised that simply memorising the location of either end of the bar (90° or 270°) will result in the correct response (90°), thereby making the VWM task considerably easier than when presenting stimuli that use the full space of 360 possible responses.” (P. 8, line 166 ff.) Ricker, T. J., Souza, A. S., & Vergauwe, E. (2023). Feature identity determines representation structure in working memory. Journal of experimental psychology. General, 152(10), 2925–2940. https://doi.org/10.1037/xge0001427
---------------	--

4. I think more attempts on speculating on the “why” is welcomed in the Discussion. Do the authors think it’s one (or many) of the methodological issues that caused a non-replication, or simply a false positive due to low statistical power from the original study?

Reply:	We thank the reviewer for this comment that has helped us improve our discussion. We added Table 3 to summarise key differences between the studies (p. 17, line 312 ff.). We further added the following discussion on possible reasons for the non-replicable PPC-tDCS benefits: “With the present study being a conceptual, and not a direct, replication of Wang et al. ’s (2019) study, there are few notable methodological differences between the two studies, which are summarised in Table 3. A particularly striking difference is the sample size, which was about 2.5 times bigger in the present than in the original study. The lack of a PPC-tDCS effect on VWM in the present study, which was supported by unambiguously strong Bayesian evidence, suggests that the original findings may have been false-positive results. The two samples were, however, comparable in age and their baseline VWM performance. This is important because differences in baseline WM capacity may contribute to the differences in results between the replication study and the original study. Previous studies showed that only low-performing participants benefited from anodal PPC stimulation (Hsu et al., 2014; Tseng et al., 2012).
---------------	--

However, based on the descriptive data available from the original study (see Figure 2 in Wang et al., 2019), baseline performance was comparable and, if anything, slightly lower in the present study.

As discussed above, we deliberately modified the design of the administered task to address methodological issues that we identified in the original study. Critically, these modifications should either not affect or increase the likelihood of observing PPC-tDCS effects. First, although different stimuli (triangles vs bars) were used, both the current study and original study tested the memory of orientation information. Both triangles and bars are basic two-dimensional shapes that people are familiar with. Therefore, the slight variation in the stimuli's shape per se is unlikely to cause the absence of PPC-tDCS VWM benefits. In fact, if tDCS benefits were robust and meaningful, they ideally should be generalisable to different stimuli and even paradigms. We chose triangles over bars as stimuli to increase the difficulty level of the VWM task, as the triangles allow for using the full space of possible orientations. Previous studies have indicated greater tDCS benefits for more challenging tasks, possibly due to more room for performance improvement (Jones & Berryhill, 2012; Papazova et al., 2020; Wu et al., 2014). Hence, if anything, a more difficult VWM task design is more likely to lead to greater tDCS benefits. Yet, we did not replicate the benefits of PPC stimulation. Second, another key difference in the VWM task design is that set size conditions were intermixed in the current study but presented in blocks in the original study. Being able to focus on only one set size condition per block, participants from the original study may have been more likely to develop effective encoding strategies, especially at higher set sizes (see Wang et al., 2020).

Furthermore, there are a few more minor differences in procedural details and the tDCS device settings that are unlikely to explain the lack of PPC-tDCS benefits relative to the original study. First, unlike the original study where participants immediately completed the VWM task after stimulation, participants in the current study completed short questionnaires (well under 5 minutes) after stimulation. This procedure is designed to measure the safety of tDCS and exclude possible adverse tDCS effects impacting the observed tDCS effects, which is a requirement for ethical approval at our institution. However, effects of tDCS of more than 10 mins typically last longer than an hour (Agboada et al., 2019; Nitsche & Paulus, 2001; Reinhart & Woodman, 2014). Hence, the short interruption by the questionnaires is highly unlikely to impact the post-stimulation effects on VWM performance. Second, the tDCS devices used in the present and the original study differed in their apparatus. Both types of tDCS devices were widely used in previously studies (TCT: Andrade et al., 2018; Dawood et al., 2019; Santos et al., 2018; NeuroConn: Das et al., 2019; Lu et al., 2019; Westwood & Romani, 2018). However, the devices differ in their ramp-up (30s vs 20s) and ramp-down (2s vs 20s) time, which are used to mimic cutaneous sensations that are associated with changing current, and thus to provide good control of condition blindness (Fonteneau et al., 2019; Thair et al., 2017). There is no consensus in the literature regarding ramp-up and ramp-down settings and our questionnaire findings confirmed a good level of condition blindness(see Supplementary Materials). ” (p. 16, line 299 ff.)

5. In the Discussion, the authors also challenges other PPC tDCS studies such as Tseng et al (2012) and Hsu et al. (2014), but it seems that several aspects of the current study are quite different from those studies (e.g., how to explain their EEG findings; low vs. high performing individual differences in tDCS receptivity; color location task) need to be accounted for?

Reply:	We thank the reviewer for this suggestion to help us strengthen our argument. We agree that several key aspects (capacity baseline; colour vs orientation WM task) are not comparable between the present study and those by Tseng et al (2012) and Hsu et al. (2014). Therefore, it would be more appropriate to remove these two citations here. “ Like Dumont et al. (2021), our findings were largely supported by substantial to strong Bayesian evidence, challenging previously reported positive effects of anodal PPC-tDCS on WM capacity (Wang et al., 2019).” (p.19, line 359)
---------------	---

6th Dec 23

Dear Ms Jiang,

Your manuscript titled "Does transcranial direct current stimulation enhance visual working memory? A replication study" has now been seen by our reviewers, whose comments appear below. In light of their advice I am delighted to say that we are happy, in principle, to publish a suitably revised version in Communications Psychology under the open access CC BY license (Creative Commons Attribution v4.0 International License).

We therefore invite you to revise your paper one last time to address the remaining concerns of our reviewers and a list of editorial requests. At the same time we ask that you edit your manuscript to comply with our format requirements and to maximise the accessibility and therefore the impact of your work.

EDITORIAL REQUESTS:

Please review our specific editorial comments and requests regarding your manuscript in the attached "Editorial Requests Table". Please outline your response to each request in the right hand column. Please upload the completed table with your manuscript files as a Related Manuscript file. Please also submit a response to the reviewers' final comments.

SUBMISSION INFORMATION:

OPEN ACCESS:

Communications Psychology is a fully open access journal. Articles are made freely accessible on publication under a CC BY license (Creative Commons Attribution 4.0 International License). This license allows maximum dissemination and re-use of open access materials and is preferred by many research funding bodies.

For further information about article processing charges, open access funding, and advice and support from Nature Research, please visit <https://www.nature.com/commspsychol/article-processing-charges>

At acceptance, you will be provided with instructions for completing this CC BY license on behalf of all authors. This grants us the necessary permissions to publish your paper. Additionally, you will be asked to declare that all required third party permissions have been obtained, and to provide billing information in order to pay the article-processing charge (APC).

* TRANSPARENT PEER REVIEW: Communications Psychology uses a transparent peer review system. On author request, confidential information and data can be removed from the published reviewer reports and rebuttal letters prior to publication. If you are concerned about the release of confidential data, please let us know specifically what information you would like to have removed. Please note that we cannot incorporate redactions for any other reasons.

* CODE AVAILABILITY: All Communications Psychology manuscripts must include a section titled "Code Availability" at the end of the methods section. We require that the custom analysis code supporting your conclusions is made available in a publicly accessible repository at this stage; please choose a repository that generates a digital object identifier (DOI) for the code; the link to the repository and the DOI must be included in the Code Availability statement. Publication as Supplementary Information will not suffice.

* DATA AVAILABILITY:

[link redacted]

Best regards,

Antonia Eisenkoeck

Antonia Eisenkoeck
Senior Editor
Communications Psychology

REVIEWERS' EXPERTISE:

Reviewer #1

Reviewer #2

REVIEWERS' COMMENTS:

Reviewer #1 (Remarks to the Author):

My apologies (a second time) for the slow response. My initial review was positive and I am satisfied with the authors' responses to my suggestions. I think the MS is improved by the responses to the other reviewers.

I have no further comments.

Reviewer #3 (Remarks to the Author):

I appreciate the changes that the authors have made to address my comments. I only have 2 points remaining:

Line 44: saying

I think the statement that this study “challenges previously reported benefits of anodal PPC-tDCS on VWM” is a bit too strong. We have established that there are many differences amongst the studies. Perhaps the statement can be more nuanced than its current form.

Line 310: can the authors provide a table of actual descriptives, instead of delta values? In the same vein, please change the values from Table 1 into actual tDCS and sham K values, and let the readers figure out the differences themselves. Right now the capacity level is claimed to be the same (or slightly lower here) but there is no reference point. Additionally, providing K values here in table format will also make it easier to compare between studies.

REVIEWERS' COMMENTS:

Reviewer #1 (Remarks to the Author):

My apologies (a second time) for the slow response. My initial review was positive and I am satisfied with the authors' responses to my suggestions. I think the MS is improved by the responses to the other reviewers.

I have no further comments.

Reply:	We thank the reviewer for reviewing the manuscript for a second time and their positive evaluation of our revision.
---------------	---

Reviewer #3 (Remarks to the Author):

I appreciate the changes that the authors have made to address my comments. I only have 2 points remaining:

Line 44: saying

I think the statement that this study “challenges previously reported benefits of anodal PPC-tDCS on VWM” is a bit too strong. We have established that there are many differences amongst the studies. Perhaps the statement can be more nuanced than its current form.

Reply:	We thank the reviewer for this helpful comment. We have toned it down as: “Therefore, our results challenge previously reported benefits of single-session anodal PPC-tDCS on VWM.” (p. 3 line 53)
---------------	--

Line 310: can the authors provide a table of actual descriptives, instead of delta values? In the same vein, please change the values from Table 1 into actual tDCS and sham K values, and let the readers figure out the differences themselves. Right now the capacity level is claimed to be the same (or slightly lower here) but there is no reference point. Additionally, providing K values here in table format will also make it easier to compare between studies.

Reply:	We thank the reviewer for this useful comment. The analyses we ran used the difference values relative to sham. Therefore, we felt that reporting the descriptive statistics (Table 2, p. 34, line 745 ff.) of the difference values is more consistent and easier for the reader to follow. In addition, we now report the actual K and SD ⁻¹ after each stimulation (sham, PPC-tDCS, and DLPFC-tDCS) in Appendix B, Table C (p .40, line 768) .
---------------	---